



# Why does stratospheric aerosol forcing strongly cool the warm pool?

Moritz Günther[1], Hauke Schmidt[1], Claudia Timmreck[1], and Matthew Toohey[2]

[1]Max Planck Institute for Meteorology, Hamburg, Germany
[2]Institute of Space and Atmospheric Studies, University of Saskatchewan, Saskatoon, Saskatchewan, Canada

**Correspondence:** Moritz Günther (moritz.guenther@mpimet.mpg.de)

**Abstract.** Previous research has shown that stratospheric aerosols cause only a small temperature change per unit forcing because they produce stronger cooling in the tropical Indian and Western Pacific Ocean than in the global mean. The enhanced temperature change in this so-called "warm pool" region activates strongly negative local and remote feedbacks, which dampen the global mean temperature response. This paper addresses the question why stratospheric aerosol forcing affects warm pool temperatures more strongly than $CO_2$ forcing, using idealized MPI-ESM simulations. We show that the aerosol's enhanced effective forcing at the top of the atmosphere (TOA) over the warm pool contributes to the warm pool-intensified temperature change, but is not sufficient to explain the effect. Instead, the pattern of surface effective forcing, which is substantially different from the effective forcing at the TOA, is more closely linked to the temperature pattern. Independent of surface temperature changes the aerosol heats the tropical stratosphere, accelerating the Brewer-Dobson circulation. The intensified Brewer-Dobson circulation exports additional energy from the tropics to the extratropics, which leads to a particularly strong negative forcing at the tropical surface. These results show how forced circulation changes can affect the climate response by altering the surface forcing pattern. Furthermore, they indicate that the established approach of diagnosing effective forcing at the TOA is useful for global means, but a surface perspective on the forcing must be adopted to understand the evolution of temperature patterns.

## 1 Introduction

Stratospheric sulfate aerosol forcing can arise naturally from volcanic eruptions, or artificially from deliberate injection of sulfur into the stratosphere. The aerosol increases reflection of shortwave (SW) radiation, which constitutes a negative forcing and cools the Earth. Sulfate aerosol also absorbs near-infrared and terrestrial longwave (LW) radiation, causing a smaller positive forcing and radiative heating in the stratosphere. Radiative forcing from stratospheric aerosol produces stronger feedback and hence a smaller temperature change per unit forcing than radiative forcing from $CO_2$ (e.g. Hansen et al., 2005; Gregory et al., 2016; Zhao et al., 2021; Günther et al., 2022). The pronounced negative feedback to volcanic eruptions contributes to variations in Earth's radiative feedback parameter over the historical period, where high volcanic activity coincides with strong global-mean feedback (Gregory and Andrews, 2016; Gregory et al., 2020; Salvi et al., 2023).

Modelling studies have shown that the strong feedback to stratospheric aerosol forcing arises from enhanced changes in warm pool (WP) temperatures relative to the global mean (Günther et al., 2022). The WP comprises the equatorial Indian and Western Pacific Ocean (30°S - 30°N, 50°E - 160°W) and is the main region of deep convection due to its high sea surface temperature (SST). The amplified temperature change in the WP increases the tropical to mid-latitude inversion strength





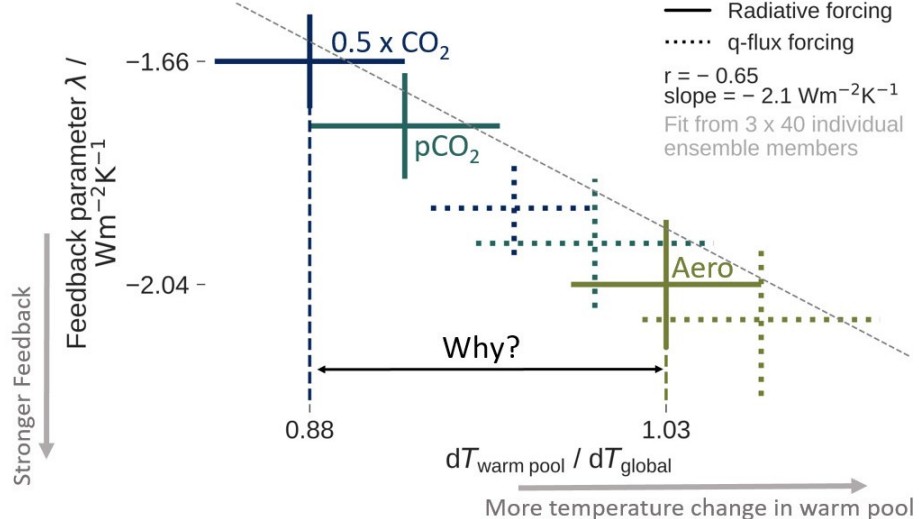

**Figure 1.** Relationship between the feedback parameter and the relative WP temperature change. The solid lines show mean and standard error from the coupled simulation ensembles. Stratospheric aerosol forcing (Aero) causes stronger WP temperature change and hence stronger feedback than $0.5 \times CO_2$ forcing. The gray dashed fit line is calculated from a regression through a total of 120 radiatively forced simulations, 40 for each forcing agent (see section 2). The results from the q-flux forced simulations are shown in dotted lines and are discussed in section 3.5.

and activates strong negative lapse rate and cloud feedbacks (Ceppi and Gregory, 2019). The cloud and lapse rate feedback processes that originate from the WP temperature change are powerful enough to impact the global-mean radiative feedback, which explains how the pronounced WP cooling from stratospheric aerosol can cause substantially more negative feedback than $CO_2$ forcing.

However, it has remained unclear why aerosol forcing impacts WP temperatures more strongly than $CO_2$ forcing, which stands as the principal incentive for pursuing this study (see Fig. 1). We explore hypotheses that could explain the causes of the different temperature patterns.

The most obvious hypothesis is that the different top of the atmosphere (TOA) effective forcing patterns cause the temperature pattern differences. Model studies have focused on the impact of aerosol from large tropical eruptions, which lead to aerosol optical depths that are largest in the low latitudes. Since both aerosol optical depth and incoming solar radiation peak in the tropics, they could combine to produce intensified low-latitude radiative forcing. In comparison, $CO_2$ forcing is relatively spatially uniform. The tropically enhanced forcing pattern from aerosols has been speculated to be the reason for the pronounced temperature changes in the tropics, in particular in the WP (Salvi et al., 2023; Günther et al., 2022).

Alternatively, the different temperature patterns of aerosol and $CO_2$ forcing could originate from other distinctive features of the forcing agents. It has been speculated that spectral differences could play a role, since aerosol forcing acts predominantly SW radiation, while $CO_2$ exclusively affects LW radiation (Joshi and Shine, 2003; Bony et al., 2006). Günther et al. (2022)



also speculated about a fundamental difference in feedback strength to positive vs. negative forcing, however, an extension of the ensemble analysed in their study made this hypothesis less plausible.

Another essential discrepancy between aerosol and $CO_2$ forcing is the heating of the stratosphere and upper troposphere due to the aerosols' absorption of radiation. The diabatic heating can alter the energy balance and the meridional temperature gradient in the upper troposphere and stratosphere. This has consequences for the strength and position of the polar vortex (e.g. Toohey et al., 2014; Azoulay et al., 2021; Bittner et al., 2016; Graf et al., 2007), and can lead to an acceleration of the Brewer-Dobson circulation (BDC), although different studies yield conflicting results (Garfinkel et al., 2017). Within the wave-

driven BDC, air moves upward in the tropical stratosphere. In the tropics, forced upwelling leads to an adiabatic cooling of the environment that depends on the vertical velocity and the temperature gradient (Birner and Charlesworth, 2017). The air then moves polewards and descends in the extratropical stratosphere where it causes adiabatic heating (Holton et al., 1995). Changes to the BDC due to stratospheric aerosol forcing have been the subject of previous research (e.g. Garfinkel et al., 2017; SPARC, 2022; Diallo et al., 2017), but the consequences for radiative feedback and temperature patterns have not been explored yet.

Motivated by the temperature pattern's importance for radiative feedbacks, we investigate which of the distinctions between $CO_2$ and aerosol forcing cause the differences in the temperature change patterns, particularly with respect to the WP. Using coupled climate model simulations, we present arguments that the pattern of TOA effective forcing is only weakly related to the pattern of surface temperatures and the radiative feedback. Instead, the surface forcing is more relevant for explaining the temperature pattern. We show that the contrast between adiabatic cooling in the tropical stratosphere and heating in the

extratropical stratosphere from an accelerated BDC causes additional negative forcing at the tropical surface, which contributes to enhanced cooling of the tropics.

## 2   Simulations and methods

### 2.1   Model

We perform simulations with the climate model MPI-ESM 1.2 in the low-resolution setup (Mauritsen et al., 2019). The atmo-

sphere component ECHAM6 (Stevens et al., 2013) is resolved with 1.875° x 1.875° at 47 levels. It is coupled to the ocean component MPIOM (Jungclaus et al., 2013), which runs on a bipolar grid with a resolution of 1.5° near the equator. MPI-ESM also includes modules for land processes and ocean biogeochemistry (Reick et al., 2021; Ilyina et al., 2013). Since no interactive atmospheric chemistry processes are included, aerosols and trace gases are prescribed with monthly climatological fields that represent unforced pre-industrial conditions.

### 2.2   Simulations

We perform simulations with three forcings: An abrupt halving of the $CO_2$ concentration ("$0.5 \times CO_2$"), an abrupt increase of the stratospheric aerosol concentration ("Aero"), and a patterned $CO_2$ simulation with spatially and seasonally varying $CO_2$ concentrations ("pCO$_2$").



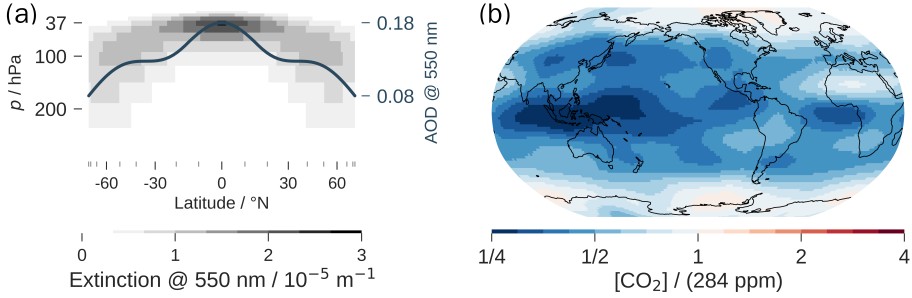

**Figure 2.** Forcing input for the simulations. (a) Aerosol radiative properties that serve as input for the Aero simulation: Extinction as function of latitude and pressure, and aerosol optical depth (AOD, vertically integrated extinction) as function of latitude, computed with EVA (Toohey et al., 2016). (b) Annual-mean field of $CO_2$ concentrations (in units of the pre-industrial $CO_2$ concentration) that serves as input for the $pCO_2$ simulation. In the actual simulation, monthly varying fields are used (appendix, Figs. A1 and A2). The fields were computed with the algorithm described in appendix A.

The Aero simulations are designed to represent the time-mean forcing induced by a strongly idealized tropical volcanic
eruption, or by deliberate stratospheric aerosol injection. We derive monthly and zonal mean fields of aerosol optical properties from the EVA forcing generator (Toohey et al., 2016) for one January and one July eruption, both with an injection mass of 20 Tg sulfur. The July eruption is then shifted by 6 months, and the average of both phase-matched eruptions is computed, in order to remove seasonal transport asymmetries while preserving a realistic poleward mass transport. We prescribe the average of the first three post-eruption years as time-invariant forcing to MPI-ESM. Constructing the aerosol forcing to be step-like in
time allows for a consistent comparison to the 0.5 x $CO_2$ forcing. The aerosol is only coupled to the radiation, not transported by the model, does not evolve in time, nor does it interact directly with clouds or ozone. While these restrictions certainly limit realism, they allow us to isolate the effects of stratospheric aerosol in an idealized, interpretable framework. The most important radiative properties of the aerosol input are shown in Fig. 2 (a).

In $pCO_2$, $CO_2$ concentrations at each grid box and month are chosen such that they give rise to an effective TOA forcing
field which is approximately equal to the TOA radiative forcing of Aero in space and time. The rationale for this experiment's design is as follows: If the WP-enhanced TOA forcing pattern of stratospheric aerosol is responsible for the WP-enhanced temperature pattern, then the same effect should appear in a $CO_2$-forced simulation with WP-enhanced forcing pattern. The iterative process that was used to determine the $CO_2$ concentrations is described in appendix A. The annual-mean input field of spatially varying $CO_2$ concentrations is shown in Fig. 2 (b). The $CO_2$ concentrations are lowest over the WP, and slightly
higher than the pre-industrial value over the poles. The resulting field of effective forcing shares these broad features and is shown in Fig. 3 (a).

To test the hypothesis that the effective forcing pattern from stratospheric aerosol causes the enhanced WP temperature change, we perform the three sets of simulations summarized in Table 1.




**Table 1.** Overview over the MPI-ESM simulations

|  | $0.5 \times CO_2$ | Aero | $pCO_2$ | Aero (non-absorbing) |
|---|---|---|---|---|
| Coupled: radiative forcing | 40 x 10 years | 40 x 10 years | 40 x 10 years | — |
| Fixed SST: radiative forcing | 100 years | 100 years | 100 years | 100 years |
| Coupled: q-flux forcing | 40 x 10 years | 40 x 10 years | 40 x 10 years | — |

### 2.2.1 Coupled simulations with radiative forcing

From a 1000-year control simulation with pre-industrial conditions (piControl), we branch one simulation for each forcing $(0.5 \times CO_2, \text{Aero}, pCO_2)$ every 25 years, leading to a total of 3 x 40 ensemble members, each run for 10 years.

### 2.2.2 Simulations with fixed SST and sea ice

As an analog to the coupled simulations, for each forcing we perform one 100-years simulation with SST and sea ice concentrations fixed to climatological control values. By subtracting the mean climate state in these perturbed simulations from
the model's mean control climate state (piClim-control), we can diagnose effective forcing at the TOA, at the surface, and adjustments (Forster et al., 2016; Sherwood et al., 2015). Results from the fixed SST simulations are averaged over all 100 simulated years except the first to allow for rapid adjustments. Forster et al. (2016) recommend 30 years to reliably diagnose the global mean effective forcing. We find that 100 years are necessary to determine also the spatial pattern of the effective forcing, especially at the surface, where interannual variability is strong.

In addition to the fixed SST simulation with the Aero forcing, we perform a simulation with non-absorbing aerosol forcing and only with fixed SST and sea ice, in order to isolate the effects that arise from the stratospheric heating, in particular the acceleration of the BDC (section 3.3). The focus will be on the absorbing aerosol forcing (Aero).

### 2.2.3 Coupled simulations with q-flux forcing

Forcing the climate system not radiatively but with a "ghost forcing" (Hansen et al., 1997) at the surface allows for an exam-
ination of the way the surface forcing pattern affects the temperature pattern, without any perturbations to the atmosphere's radiative properties. We derive the surface effective forcing from the fixed SST simulations as the difference between all surface fluxes (radiative and turbulent) of the perturbed simulations and piClim-control. We then prescribe these flux anomalies as an additional heat source / sink ("q-flux") to the ocean and compute an ensemble of 40 simulations for each forcing agent, where each simulation lasts for 10 years. Note that the atmosphere is still fully coupled to the dynamical ocean.

## 2.3 CMIP6 output

We complement the dedicated MPI-ESM simulations with output of the piControl and historical simulations from phase 6 of the Coupled Model Intercomparison Project (CMIP6; Eyring et al., 2016) to test the results on other models. We include all 23





models that provide the necessary output to compute adiabatic cooling in the stratosphere according to Eq. 2 (see section 2.4.4). For models with multiple realizations the ensemble-mean is calculated after applying Eq. 2, so that each model is weighted equally. A list of all models and the number of ensemble members is found in Table C1 in the appendix.

## 2.4 Defining forcing, feedback, WP-enhancement, and adiabatic cooling

### 2.4.1 Effective forcing

Effective forcing is defined as the time-mean flux change in a perturbed simulation compared to an unperturbed control simulation, both with the same prescribed SST and sea ice (Forster et al., 2016). It is traditionally measured at the TOA, where it consists of SW and LW flux changes. We also diagnose effective forcing at the surface, where additionally the sensible and latent heat fluxes must be taken into account.

### 2.4.2 Feedback parameter

We employ the definition of the "differential feedback parameter" following Rugenstein and Armour (2021) as $\lambda = \frac{\partial N}{\partial T}$ with global-mean TOA flux $N$ and global-mean near-surface air temperature $T$, obtained by regression over ten years.

### 2.4.3 Warm pool Index (WPI)

Given the elevated role of the WP, spatial patterns can be meaningfully measured with a simple WP index (WPI), which indicates how strongly a quantity is concentrated in the WP. For patterns of effective forcing $F$, we define it as $\mathrm{WPI}_F = F_{\mathrm{WP}}/F_{\mathrm{global}}$. Temperatures vary with time, so for patterns of temperature change $T$ we define $\mathrm{WPI}_T = \mathrm{d}T_{\mathrm{WP}}/\mathrm{d}T_{\mathrm{global}}$, obtained by regression over 10 years. Values greater than one indicate greater forcing or temperature change in the WP than in the global mean.

### 2.4.4 Adiabatic cooling in the stratosphere

Upwelling in the tropical stratosphere causes adiabatic cooling of rate $K$ (in $\mathrm{Ks}^{-1}$), which is proportional to the upwelling speed $w$ and the deviation of the temperature profile $\frac{\partial T}{\partial z}$ from a dry adiabat $-\frac{g}{c_p}$ (Birner and Charlesworth, 2017):

$$K = -w \left( \frac{\partial T}{\partial z} + \frac{g}{c_p} \right) \tag{1}$$

with the specific heat capacity of dry air $c_p$ and gravitational acceleration $g$. Using the hydrostatic approximation $\rho dz = -dp/g$, we calculate the integrated adiabatic cooling of the stratosphere as power flux density $Q_{\mathrm{adi}}$ (in $\mathrm{Wm}^{-2}$) according to



$$Q_{\mathrm{adi}} = c_p \int\limits_{\mathrm{strat.}} K(z)\rho(z)\,dz$$

$$= -\frac{c_p}{g} \int\limits_{100\,\mathrm{hPa}}^{1\,\mathrm{hPa}} K(p)\,dp \tag{2}$$

By vertically integrating over the stratosphere we effectively treat it as one layer that causes adiabatic cooling. Changing the
lower limit to 70 hPa changes the numbers by up to 25 %, but not in a way that would affect the conclusions.

The changes in $K$ can be decomposed linearly into contributions from changes in $w$ and $\frac{\partial T}{\partial z}$:

$$\Delta_w K = \frac{\partial K}{\partial w}\bigg|_0 \Delta w \quad = -\Big(\frac{\partial T}{\partial z}\Big|_0 + \frac{g}{c_p}\Big)\Delta w \tag{3}$$

$$\Delta_{\frac{\partial T}{\partial z}} K = \frac{\partial K}{\partial \frac{\partial T}{\partial z}}\bigg|_0 \Delta \frac{\partial T}{\partial z} = -w\big|_0 \Delta \frac{\partial T}{\partial z} \tag{4}$$

The notation $|_0$ indicates values in the unperturbed state, i.e. from the piClim-control or piControl simulation. Plugging these
cooling rates into Eq. 2 yields the adiabatic cooling changes $\Delta Q_{\mathrm{adi}}$ due to changes in $w$ and $\frac{\partial T}{\partial z}$.

## 3   Results

### 3.1   Effects of the TOA forcing pattern on the temperature change pattern

First, we explore the hypothesis that the WP-enhanced forcing pattern from volcanic aerosol causes the WP-enhanced temper-
ature response. The TOA effective forcing fields of the radiatively forced simulations are shown in Fig. 3 (a). All fields average
globally to approximately $-3.5\ \mathrm{Wm}^{-2}$, but exhibit different patterns. Compared to the relatively uniform $0.5 \times CO_2$ TOA
effective forcing pattern, Aero exhibits enhanced negative forcing over the tropics, in particular over the WP. The forcing over
Antarctica is slightly positive because the aerosols' SW effect is diminished due to low insolation, high surface albedo, and
possibly because of adjustments, while the LW effect remains. By design, the $pCO_2$ TOA forcing field shares the Aero forcing
field's main features, although it is slightly less enhanced over the WP, which we will address later. The TOA effective forcing
fields of Aero and $pCO_2$ agree well not only in the annual mean, but also in each month (appendix, Figs. A1 and A2).

The temperature change patterns of all coupled 10-year simulations are broadly similar (Fig. 3 (b)). Temperature change
is amplified in the Arctic, moderate in low latitudes including the WP, and suppressed over the Southern Ocean. The most
relevant region for the feedback is the WP, where small differences have substantial impacts for global feedback, with changes
of roughly $-0.2\ \mathrm{Wm}^{-2}\mathrm{K}^{-1}$ per 0.1 $\mathrm{WPI}_T$ points in MPI-ESM (Fig. 1). The definition $\mathrm{WPI}_T = dT_{\mathrm{WP}}/dT_{\mathrm{global}}$ is equivalent
to the WP average of the temperature pattern shown in the black box in Fig. 3 (b). Despite the fact that the temperature pattern
differences among the simulations over the WP are relatively small compared to those in other regions, these small changes
dominate the global mean radiative feedback parameter (Dong et al., 2019; Günther et al., 2022).





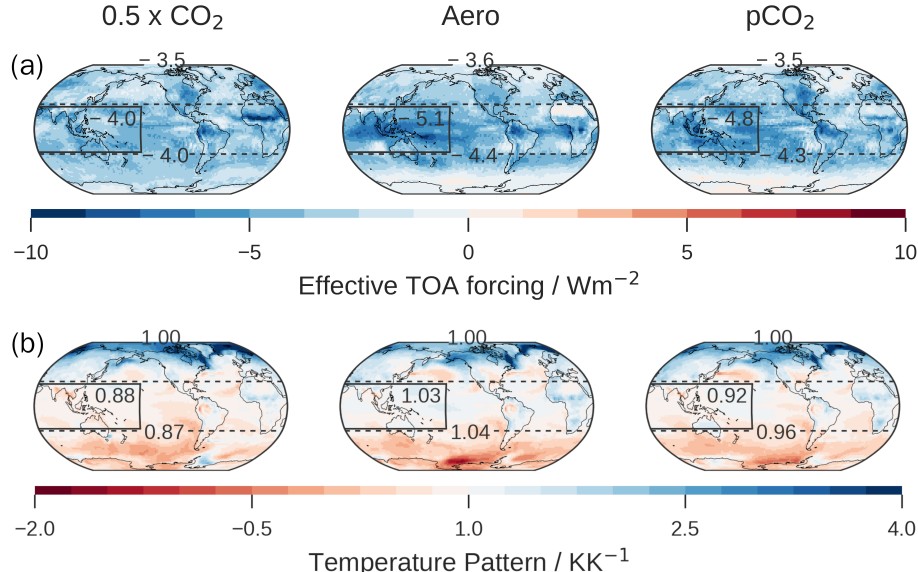

**Figure 3.** (a) Annual-mean TOA effective forcing obtained from the difference between the forced simulations with fixed SST and the control simulation with fixed SST. (b) Ensemble-mean 2-meter temperature change patterns (ratio of local to global mean temperature change) of the radiatively forced, coupled simulations. Values greater than one indicate stronger than global average cooling, lower than one weaker than average cooling. Values lower than zero indicate warming. WP and tropics are shown with solid and dashed lines, and the field average over these regions is shown in the WP box and in the dashed line, respectively. The global mean is shown at the top of each panel.

Although Aero and $pCO_2$ are almost equally strongly forced in the WP, the WPI of the temperature pattern of $pCO_2$ (0.92) is smaller than that of Aero (1.03). We perform Student's t-tests on the distributions of $WPI_T$ from the 40 ensemble members

of each forcing agent under the null hypothesis that they are drawn from distributions with the same average. While the $WPI_T$ values of $0.5 \times CO_2$ and $pCO_2$ are not significantly different (p = 0.2), the $WPI_T$ values of Aero and $pCO_2$ are distinct (p = $10^{-6}$). Although the TOA forcing patterns of $pCO_2$ and Aero are similar, their temperature change patterns are not, when measured by the $WPI_T$ (Fig. 1). This contradicts the hypothesis that the WP-enhanced TOA forcing pattern of Aero causes the WP-enhanced temperature change pattern. If that were the case, the WP should cool equally strongly in Aero and $pCO_2$ .

This result is limited by the fact that the forcing pattern of $pCO_2$ is not quite as WP-enhanced as in Aero: The $WPI_F$ (WP forcing divided by global mean forcing) is only 1.36 in $pCO_2$, but 1.44 in Aero. However, even when applying a correction factor of 1.44/1.36 to the $WPI_T$ values of $pCO_2$, the $WPI_T$ values remain significantly different from Aero, albeit with higher p-value (p = 0.02).

In summary, the $pCO_2$ simulation with a TOA forcing pattern almost as WP-heavy as Aero, does not produce a temperature

change pattern as WP-heavy as Aero. Instead, its temperature change pattern is rather similar to the $0.5 \times CO_2$ simulation (see also Fig. 1). Hence, we arrive at the conclusion that another process which is specific to aerosol forcing must cause the temperature pattern differences.



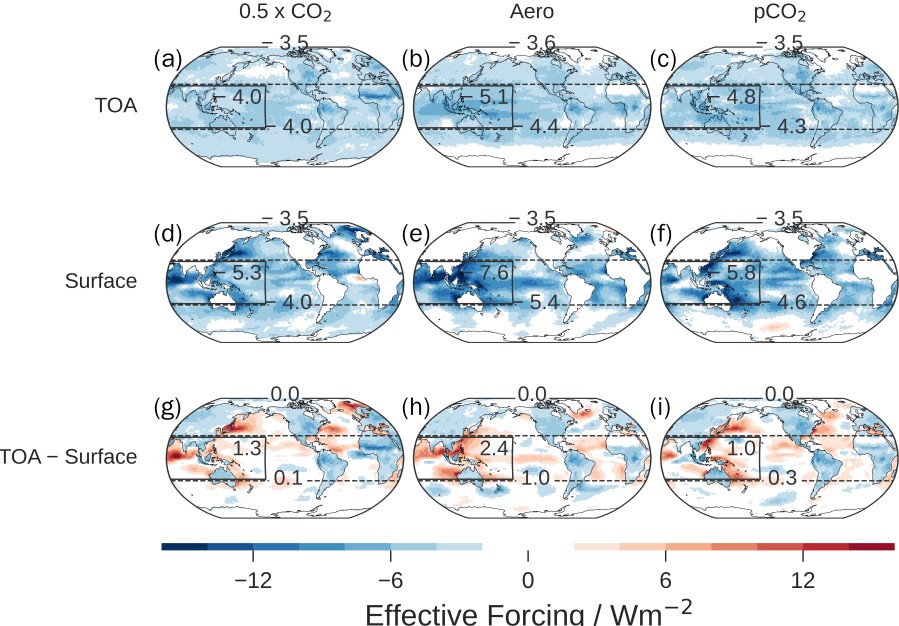

**Figure 4.** Effective forcing of the radiatively forced simulations at TOA (row 1), surface (row 2), and their difference (row 3). The difference between TOA and surface effective forcing is the effective forcing on the atmosphere, and can be interpreted as a redistribution of negative forcing from blue to red regions, when going from the TOA to the surface. WP and tropics are shown with solid and dashed lines, and the field average over these regions is shown in the WP box and in the dashed line, respectively. The global mean is shown at the top of each panel.

## 3.2 Surface forcing pattern

The rationale of the TOA-forcing hypothesis was that stronger forcing in the WP could lead to stronger temperature change in
the WP. However, the ocean does not directly respond to the forcing at the TOA, but the forcing at the surface, which might therefore be more relevant. In the following section, we examine the surface effective forcing and how it differs from the TOA effective forcing.

There are multiple constraints to surface forcing: land and atmosphere have small heat capacities and therefore cannot act as energy reservoirs on time scales of the 100 years fixed SST simulation. If there were substantial fluxes into the atmosphere or
the land, they would heat up or cool down until the fluxes become zero. The time scale of these adjustments is fast due to the small heat capacities of land and atmosphere. Only the ocean (due to its fixed SST) and the TOA can support non-zero fluxes in the steady state of the fixed SST simulations. These considerations imply that the surface effective forcing over land must be zero. The fact that the global-mean flux into the atmosphere is zero implies that the global-mean effective forcings at the TOA and the surface must be equal (see also Eq. 5).





Fig. 4 shows the annual mean effective radiative forcing at the TOA, the surface, and their difference, diagnosed from the fixed SST simulations. The surface forcing exhibits a richer spatial structure than the TOA forcing. Furthermore, the WP-intensification of the forcing in Aero is much more pronounced at the surface than at the TOA: At the WP surface, it is more than twice as strong as in the global mean. In comparison, the surface forcing in $pCO_2$ is only slightly more WP-enhanced than in $0.5 \times CO_2$.

The difference between TOA and surface forcing is the effective forcing that acts on the atmosphere, which is zero in the global mean, but has a pattern. It must be locally balanced by changes in the horizontal heat flux divergence $Q$, since the atmosphere has no relevant sinks or sources of energy on long time scales.

$$F_{\text{TOA}} - F_{\text{surface}} = F_{\text{atm}} \tag{5}$$

$$F_{\text{atm}} + Q = 0 \tag{6}$$

$F_{\text{TOA}}$ and $F_{\text{surface}}$ denote the effective forcing *at* the TOA or *at* the surface, respectively. $F_{\text{atm}}$ denotes the effective forcing *on* the atmosphere. All effective forcing fields are a function of longitude and latitude. Both sides of Eq. 5 globally average to zero.

Alternatively, the forcing on the atmosphere can be interpreted as an atmospheric redistribution of forcing, or simply as the change in horizontal atmospheric heat flux divergence at fixed SST. Taking a perspective from the TOA looking down

to the surface, the atmosphere shifts negative forcing from grid points with negative values towards grid points with positive values. Negative forcing is redistributed from columns over land to columns over ocean in all simulations (Fig. 4, bottom row). This effect arises somewhat artificially from the fact that effective forcing is diagnosed at fixed SST concentrations and sea ice concentrations, but not fixed land temperatures. In Aero there is an additional convergence of negative forcing at the WP surface ($2.4 \text{ Wm}^{-2}$ compared to $1.0 \text{ Wm}^{-2}$ in $pCO_2$ and $1.3 \text{ Wm}^{-2}$ in $0.5 \times CO_2$).

We argue that the surface forcing is the critical factor that distinguishes aerosol from $CO_2$ forcing. The differences between TOA and surface forcing are imposed by heat transport changes that arise from anomalous circulations. They come about as adjustments which are specific to the forcing agent. We hypothesize that the WP surface in Aero is so strongly forced because the anomalous atmospheric circulation leads to an anomalous energy transport out of the WP, or, equivalently, moves positive forcing away from the WP surface. In the following sections, we aim to (1) explain how the differences in atmospheric

forcing divergence arise from the circulation changes, and (2) determine if these differences cause the distinctions among the temperature change patterns.

### 3.3   Explaining the atmospheric forcing divergence

What explains the different structures of the forcing on the atmosphere (bottom row of Fig. 4)? Most of the spatial structure arises from variations in the latent heat flux at the surface (not shown). Therefore, it is mostly the latent heat flux that re-

acts to energetic constraints from the atmosphere, consistent with previous studies (Fajber and Kushner, 2021; Fajber et al.,




**Table 2.** Adiabatic cooling in the stratosphere calculated according to Eq. 2 and averaged over the tropics (30°N to 30°S). The lower rows show the contributions from changes in the upwelling speed and the lapse rate, respectively.

|  | $0.5 \times CO_2$ | Aero | pCO$_2$ | Aero (non-absorbing) |
|---|---|---|---|---|
| $\Delta Q_{\mathrm{adi}}$ / Wm$^{-2}$ | 0.0 | -0.8 | -0.3 | 0.0 |
| $\Delta Q_{\mathrm{adi}}$ due to $\Delta w$ / Wm$^{-2}$ | 0.1 | -0.7 | -0.2 | 0.0 |
| $\Delta Q_{\mathrm{adi}}$ due to $\Delta \partial T/\partial z$ / Wm$^{-2}$ | -0.1 | -0.2 | -0.2 | 0.0 |

2023). However, this does not explain why the atmosphere redistributes energy, and which circulations accomplish this energy transport.

Eq. 6 states that the forcing on the atmosphere is balanced by heat transport. We focus on the anomalous energy export out of the WP, and on the question why this anomaly is stronger for Aero. To this end we partition the energy export into a meridional component, i.e. the energy transport from the tropics to the extratropics, and a tropical-zonal component, i.e. the transport from the WP to tropical non-WP regions. In the "TOA - Surface" row of Fig. 4, the meridional energy transport is equal to the average over the tropics (1.0 Wm$^{-2}$ in Aero, compared to 0.1 Wm$^{-2}$ in $0.5 \times CO_2$) , and the zonal energy transport is measured by the difference between the WP mean and the tropical mean (1.4 Wm$^{-2}$ in Aero compared to 1.2 Wm$^{-2}$ in $0.5 \times CO_2$).

Obvious explanations for these transports could be changes in gross moist stability, the Hadley or Walker circulation, or eddy-energy flux. Indeed, the tropical tropospheric zonal overturning circulation has been shown to *weaken* as a consequence of stratospheric heating (Ferraro et al., 2014; Simpson et al., 2019). However, a weaker zonal overturning circulation cannot be the reason for increased energy flux out of the WP, unless it is overcompensated by increases in gross moist stability. We argue that the circulation changes associated to the atmospheric energy budget do not project onto the Hadley and Walker circulations. Instead, the meridional transport is accomplished via the BDC, and the tropical-zonal transport arises from an anomalous ocean-land circulation. This only applies to the direct circulation adjustments, and we do not make any statement about temperature-dependent changes to the Hadley or the Walker circulation from stratospheric aerosol forcing. The meridional and tropical-zonal component will be treated separately in the following.

### 3.3.1 Meridional energy transport: adiabatic cooling from the Brewer-Dobson circulation

We propose that the meridional transport of energy from the tropics to the extratropics arises from increased adiabatic cooling in the tropics via the BDC. Stratospheric aerosol causes a meridional heating gradient in the stratosphere, which affects the wave propagation and therefore the wave driving of the BDC. An increase in adiabatic cooling could arise from an acceleration of the vertical velocity (i.e. the BDC) or an increase of the vertical temperature gradient (see Eq. 1). The changes in adiabatic cooling in the fixed SST simulations compared to piClim-control, calculated according to Eq. 2, are shown in Table 2. In the tropics, the BDC causes an additional adiabatic cooling of 0.8 Wm$^{-2}$ in Aero, 0.0 Wm$^{-2}$ in $0.5 \times CO_2$, and 0.3 Wm$^{-2}$ in pCO$_2$. In Aero, most of this is driven by changes in the upwelling speed, and only a small part is due to changes in the





stratospheric lapse rate. While the upwelling speed influences the adiabatic cooling proportionally according to Eq. 2, changes in the temperature profile influence adiabatic cooling only in so far as they change the difference between the actual lapse rate and the dry adiabatic lapse rate. Since this difference is already quite large in the unperturbed stratosphere (10 K/km dry adiabatic lapse rate vs. -2 K/km stratospheric lapse rate), moderate changes to the stratospheric lapse rate will only have a small effect on the adiabatic cooling. However, eventually all changes to adiabatic cooling are driven by changes to the global stratospheric temperature distribution, since it is the differential heating between equator and pole that affects wave propagation and therefore the speed of the BDC.

Adiabatic cooling is not a sink of energy in the global energy budget. The energy is released during the sinking motion in the extratropics, and therefore constitutes an energy transport from the tropics to the extratropics. The mechanism of energy export due to the BDC explains the meridional energy transport of slightly less than 1 $\mathrm{Wm}^{-2}$ from tropics to extratropics seen in Fig. 4 (h). The forcing anomalies are communicated from the stratosphere to the tropical tropopause layer and the free troposphere via radiation, and then passed on via convection to the surface.

We turn to the additional simulation with fixed SST and a prescribed aerosol forcing similar to Aero, where the aerosol is modified to only scatter, but not absorb radiation. This precludes the aerosol from heating the stratosphere, which should in turn prevent the BDC mechanism. Consistent with our expectations, we find no anomalous energy transport from the tropics to the extratropics (Table 2).

Why is there a small additional adiabatic cooling in the tropics in the pCO$_2$ simulation (0.3 $\mathrm{Wm}^{-2}$)? The strongly reduced CO$_2$ concentrations in the tropics lead to a more pronounced heating of the tropical than the extratropical stratosphere. This increases both $\partial T/\partial z$ and the upwelling speed of the BDC due to the heating gradient between the equatorial and polar stratosphere. However, the stratospheric heating is not as pronounced as in Aero, and therefore the effects are smaller.

Note that an increase in adiabatic cooling does not imply that the tropical stratosphere becomes colder - it just warms less than it would if there was no acceleration of the BDC.

### 3.3.2 Tropical-zonal energy transport: Ocean-to-land circulation

All simulations exhibit a zonal energy transport from the WP to the tropical non-WP regions. This tropical-zonal energy transport is not a big contributor to the differences between the Aero and $0.5 \times$ CO$_2$ forcing patterns and therefore not a focus of our study. Nevertheless, we briefly lay out the reasons for this anomalous circulation. The explanations are somewhat rooted in model-specifics and only partially apply to the real world.

Since land temperatures vary freely in the fixed SST simulations, the land cools down rapidly, which leads to an enhanced energy flux from the atmosphere to the land. For the atmospheric energy budget to be closed, this energy must be replenished from the ocean, which can draw from an infinite energy reservoir due to the fixed SST ocean surface. Consequentially, an anomalous energy transport arises from ocean towards land (see appendix Fig. B1 and accompanying text). This is accomplished by land-to-ocean winds at the surface, and ocean-to-land winds aloft, which is also the direction of the energy flow. Since deep convection is impeded over non-WP regions, the anomalous circulation predominantly transports energy from WP





ocean regions to the tropical land regions. For this reason, there is an additional energy export from the WP to the non-WP regions in all fixed SST simulations.

Previous studies have noted the importance of land-ocean temperature contrast and a resulting monsoon-like circulation in the fast response to abrupt forcing (Modak et al., 2016; Heede et al., 2020). While this circulation arises artificially in our fixed-SST simulations from the fact that land temperatures are not fixed, it would appear similarly in reality because land reacts

much faster than ocean. This effect is important on a time scale of several months (Modak et al., 2016), which is comparable to the time scale of volcanic aerosol forcing. Our simulations show that this circulation causes an energy export out of the WP, and might therefore contribute to a strengthening of the feedback to volcanic eruptions, simply due to the time scale they act on. For long-term forcing such as anthropogenic greenhouse gases and solar radiation management, this effect would be less important. This raises the question how the different time scales of volcanic aerosol and $CO_2$ forcing affect the feedback.

Further explanations and a figure showing the anomalous circulation can be found in appendix B.

### 3.3.3  Sum of meridional and zonal terms

In total, the atmosphere exports $1.3\,\mathrm{Wm}^{-2}$ out of the WP in the $0.5 \times CO_2$ simulation, but $2.4\,\mathrm{Wm}^{-2}$ in Aero, which leads to enhanced negative forcing at the WP surface in Aero (Fig. 4). In Aero, the atmosphere absorbs $0.8\,\mathrm{Wm}^{-2}$ of positive forcing in the tropics and exports it to the extratropics via the BDC (Table 2). The energy transport via the BDC explains most of

the meridional energy transport, and the difference between the surface forcing in the Aero and $0.5 \times CO_2$ experiments. The cooling of the land surface causes an additional zonal transport of energy from the WP ocean to tropical land regions via the free troposphere in all simulations. While the anomalous meridional transport via the BDC does not show substantial year-to-year variation, the zonal energy transport varies substantially interannually. Even using 100 years of fixed SST simulations, its standard error (standard deviation in time divided by square root of sample size) is on the order of 10 to 15 %, which

is an order of magnitude higher than the standard error of the meridional transport. This implies the existence of substantial interannual variability, which originates from the atmosphere alone. Furthermore, the tropical-zonal energy transports of Aero and $0.5 \times CO_2$ differ only within one standard error, and arise at least partly from the specifics of fixed SST simulations, where temperatures are only prescribed at the ocean surface, but not at the land surface. We therefore emphasize the BDC changes as the more important result, and caution with the interpretation of the zonal energy transport. In the following, we examine the

forcing patterns and the meridional energy transport mechanism in CMIP6 models.

### 3.4  Testing the energy export mechanism with CMIP6 models

Salvi et al. (2023) provide an analysis of CMIP6 forcing patterns at the TOA and the surface (see their Fig. 10). For aerosol forcing from the Krakatau and Pinatubo eruptions, they find enhanced forcing in the tropics at the TOA and at the surface. However, while the WP-enhancement of aerosol forcing at the surface is clearly visible, it it not substantially stronger than in

the case of greenhouse gas-forcing. This calls into question the existence of the BDC mechanism in CMIP6 models.

We explicitly test if the energy export due to the BDC increases after volcanic eruptions. 23 models provide the necessary output to compute the adiabatic cooling in the historical coupled simulations. We compute the changes to the adiabatic cooling



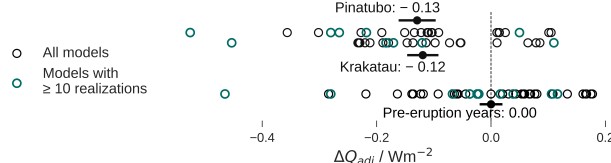

**Figure 5.** Tropical mean changes in stratospheric adiabatic cooling in the CMIP6 historical simulations in the post-eruption years of Krakatau (1884) and Pinatubo (1992) and the pre-eruption years (1882 and 1990), compared to the piControl simulations. The value from each model is shown by a circle. The green circles are from the five models with at least ten realizations. Together, these models account for 80 % of all realizations. All values are calculated using Eq. 2. The error-bars show the multi-model mean and standard error. Negative values indicate increased adiabatic cooling.

in the tropics according to Eq. 2 in the first post-eruption year of the Krakatau and Pinatubo eruption, respectively (1884 and 1992). Most models show a moderate increase in adiabatic cooling, with a multi-model average of 0.12 / 0.13 $\mathrm{Wm}^{-2}$ for the

Krakatau / Pinatubo eruption, respectively (Fig. 5). The multi-model mean +/- standard error does not overlap with zero, indicating the presence of a significant effect. The adiabatic cooling in the pre-eruption years (1882 and 1990) is indistinguishable from the control simulation, which implies that the additional adiabatic cooling in the post-eruption years is indeed caused by the eruption and not some other historical forcing agent.

The effect is smaller than what we find in the fixed SST simulations of MPI-ESM: Assuming a peak global mean effective

forcing from Krakatau and Pinatubo of approximately 1.8 $\mathrm{Wm}^{-2}$ (Salvi et al., 2023), the BDC redistributes about 7 % of the global mean forcing from the tropics to the extratropics in CMIP6, compared to about 23 % in our simulations. This difference is diminished when only taking into account the models with at least 10 realizations, where the influence of internal variability is reduced. In these models, the transport from the BDC is -0.16 $\mathrm{Wm}^{-2}$ for Krakatau and -0.25 $\mathrm{Wm}^{-2}$ for Pinatubo, corresponding to roughly 9 % and 14 % of the global mean forcing, respectively. Apart from model-differences, the remaining

disparity between our results and the CMIP6 results could arise for two systematic reasons: First, the aerosol is short-lived and not all of the post-eruption year is equally strongly affected by the presence of the aerosol. Second, the CMIP6 estimate is likely biased low, because the historical coupled simulations cool in the post-eruption year, and there is a positive correlation between temperatures and the strength of the BDC (Garfinkel et al., 2017). The last point does not compromise our finding that the BDC reshapes the surface forcing pattern, since the forcing is defined at zero surface temperature change.

In agreement with our CMIP6 analysis, model studies consistently show an acceleration of the BDC after volcanic eruptions (Garfinkel et al., 2017; Garcia et al., 2011; Pitari and Rizi, 1993; Pitari and Mancini, 2002; Aquila et al., 2013; Toohey et al., 2014; Muthers et al., 2016). In contrast, studies using reanalysis or observations provide mixed results. Some studies find enhanced wave activity in at least one hemisphere (Graf et al., 2007; Schnadt Poberaj et al., 2011), others do not find stratospheric circulation changes after volcanic eruptions (Diallo et al., 2012; Seviour et al., 2012; SPARC, 2022). Since reanalysis

products do not assimilate aerosol data and are hence ignorant to the ensuing heating rate anomalies in the stratosphere, they might not be a suitable tool to study the links between stratospheric heating and the BDC (Abalos et al., 2015). The absence



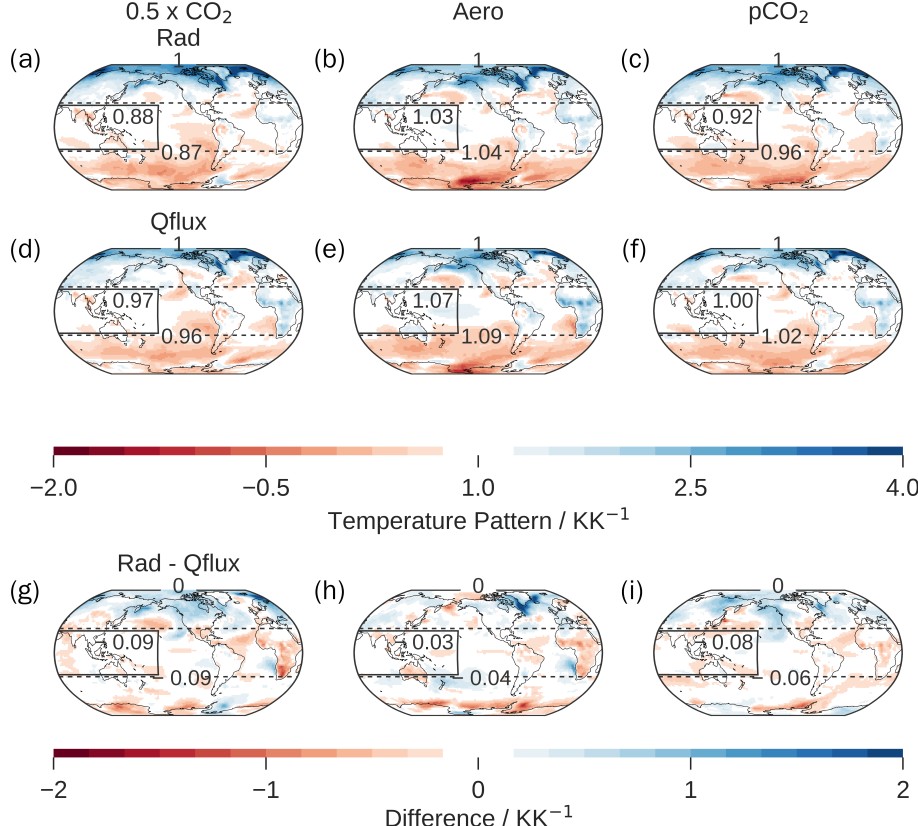

**Figure 6.** Ensemble-mean 2-meter temperature change patterns (ratio of local to global mean temperature change) of the radiatively forced simulations (a - c), q-flux-forced simulations (d - f), and their difference (g - i). In panels (a) to (f), values greater than one indicate stronger than global average cooling, lower than one weaker than average cooling. Values lower than zero indicate warming.

of the upwelling effect in observational records might also be related to the choice of the metric: Toohey et al. (2014) argue that the upwelling change might be most pronounced in the middle and upper stratosphere, while observational studies focus on upwelling in the lower stratosphere.

### 3.5 Does the surface forcing pattern cause the surface temperature pattern?

If the previously demonstrated differences in the surface forcing cause the temperature pattern differences, then these differences should appear in surface-forced simulations without any changes to aerosol or $CO_2$ concentrations. While it seems intuitive, it is not straightforward that stronger surface forcing in the WP causes stronger temperature change in the WP. Results from q-flux Green's functions have shown that the temperature response to a localized surface flux is typically non-local (Lin
et al., 2021; Liu et al., 2018a, b, 2022). We use the q-flux simulations to establish the link between local forcing and local





temperature change in the WP. They are forced by a surface heat sink / source, each of them with a global mean of -3.5 $\mathrm{Wm}^{-2}$, but with the patterns diagnosed from the radiative forcing simulations (Fig. 4 d-f).

Results from the q-flux simulations in the space of temperature pattern and feedback are shown in Fig. 1. The simulations with more strongly concentrated surface forcing in the WP also cause stronger temperature change in the WP (Aero > pCO$_2$

> 0.5 × CO$_2$). These results support our hypothesis that stronger surface forcing in the WP leads to stronger surface temperature change in the WP. However, the temperature patterns differ substantially between the q-flux forced and the corresponding radiatively forced simulations (Fig. 6), in particular with respect to temperature changes in the WP and the tropics. All q-flux forced simulations cool more strongly in the WP and in the tropics than their radiatively forced counterparts. Generally, the pattern differences between radiatively and q-flux forced simulations of the same forcing agent are in the same order of

magnitude as the differences among the forcing agents. Possible reasons for these deviations are: (i) the lack of changes to the atmospheric CO$_2$ concentrations / aerosol load, which affects the vertical structure of the atmosphere and the general circulation; (ii) the lack of forcing over land; (iii) the fact that we only include heat fluxes, but no momentum or freshwater fluxes to force the ocean surface. Aquaplanet simulations show little difference between radiatively forced and q-flux forced simulations (Haugstad et al., 2017), rendering hypothesis (i) - the absence of CO$_2$ / aerosol in the atmosphere - as a cause for

the differences unlikely. On the other hand, both CO$_2$ forced simulations show a more positive PDO-like temperature change pattern in the radiatively forced, compared to the q-flux-forced simulations, which could be an indication that this part of the temperature pattern is due to the direct effect of CO$_2$. In CESM2 simulations with historical forcing, the absence of wind stress forcing causes statistically significant changes to the SST pattern (McMonigal et al., 2023) towards a more WP-enhanced temperature pattern, consistent with the bias in our simulations and with hypothesis (iii), the momentum forcing-hypothesis.

The exact causes are beyond the scope of this study and warrant further research. Despite the shortcomings, we interpret the results from the q-flux forced simulations as a support for our hypothesis that strong surface forcing in the WP leads to strong surface cooling in the WP.

## 4   Discussion and conclusions

In this study, we identify a mechanism that redistributes radiative forcing from the tropics to the extratropics via an accel-

erated BDC due to stratospheric heating. Using MPI-ESM simulations of idealized CO$_2$ and aerosol forcing we explain the pronounced WP cooling from stratospheric aerosol forcing with the strongly negative forcing it causes at the WP surface. This finding enhances our understanding of the formation of the WP-enhanced temperature change pattern in response to strato-spheric aerosol forcing, which has previously been shown to cause strongly negative feedback (Günther et al., 2022; Salvi et al., 2023; Zhou et al., 2023).

The effective forcing from stratospheric aerosol is more negative in the WP than in the global mean already at the TOA. Furthermore, there is a substantial export of energy from the tropics to the extratropics via the stratosphere, effectively removing additional energy from the tropical surface. The stratospheric energy export emanates mainly from an acceleration of the BDC, which leads to increased adiabatic cooling in the tropical stratosphere and adiabatic heating in the extratropical stratosphere.



Changes in the BDC ultimately arise from the differential heating between the tropical and the extratropical stratosphere, which
affects wave activity and therefore the strength of the stratospheric pump (Holton et al., 1995; Graf et al., 2007; Schnadt Poberaj
et al., 2011).

The time-constant forcing we use to model stratospheric aerosol forcing is reminiscent of strategies to cool the Earth with
solar radiation management by deliberate injection of reflective aerosol into the stratosphere. Depending on the location and
absorptivity of the used aerosol the BDC will accelerate and lead to stronger cooling of the tropics than the extratropics.
Tropical overcooling is a notorious problem of solar radiation management, unless more sophisticated injection strategies are
used (Laakso et al., 2017; Kravitz et al., 2019). The importance of the BDC for the climate response corroborates the finding
from previous studies that solar dimming is an imperfect substitute for simulating aerosol forcing (Ferraro et al., 2014; Simpson
et al., 2019; Visioni et al., 2021).

In order to highlight differences between aerosol and $CO_2$ forcing independent of the forcing pattern, we created a patterned
$CO_2$ simulation, which approximately reproduces the TOA effective forcing pattern of stratospheric aerosol. This was achieved
by varying the $CO_2$ concentration in space and time. The aerosol and the patterned $CO_2$ simulation have more negative TOA
radiative forcing in the WP and the tropics than in the global mean. Despite their similar TOA effective forcing patterns,
they exhibit substantial temperature pattern differences. We therefore argue that the increased energy export out of the tropics
due to the acceleration of the BDC is essential for the emergence of the tropically enhanced temperature change pattern and
strong feedback to stratospheric aerosol forcing. This shows that the TOA forcing perspective is not sufficient to explain the
temperature patterns. The TOA perspective has previously been used as an explanation for the feedback to aerosol forcing
(Salvi et al., 2022, 2023), to solar forcing (Kaur et al., 2023; Modak et al., 2016), and for explaining temperature change
patterns in general (Liu et al., 2022). We show that the temperature pattern is more closely linked to the pattern of forcing at
the surface than at the TOA. The TOA forcing perspective is established in climate science, which is appropriate as long as the
focus is on global means, but the patterns of TOA and surface forcing can be substantially different. Differences between them
arise from changes in the atmospheric heat transport, which lead to a considerable redistribution of forcing between different
regions of the Earth. This should be kept in mind when addressing the relationship between forcing patterns and temperature
change patterns in future studies.

The comparison of radiatively forced simulations with simulations that were forced with an equivalent heat flux forcing at
the surface, reveals the existence of a link between surface forcing pattern and surface temperature response. Stronger surface
forcing in the WP produces stronger temperature change in the WP. Still, knowledge of the surface effective forcing in our
simulations is not enough to reproduce the exact temperature response.

For the interpretation of our results it should be kept in mind that the aerosol in our model has a highly idealized profile
with no seasonal dependence, and is not transported. The ozone profile is fixed, although ozone is affected by the presence
of aerosol and has been shown to affect the BDC (Garfinkel et al., 2017; Pitari and Rizi, 1993; Schnadt Poberaj et al., 2011).
Analysis with a model that permits interactive aerosol transport and chemistry could provide more insight into the magnitude
of the effects, although the contributions would be harder to disentangle.



Furthermore, shifting the aerosol profile in height or latitude would likely modify the effect on the BDC, so that our results may be dependent on the specific aerosol profile we chose. Aerosol that is injected at greater heights has been found to cause

slightly less negative feedback, but this has remained unexplained (Zhao et al., 2021). The effect could potentially be related to the height dependence of the effect of stratospheric heating on the BDC. Extratropical eruptions would not only cause a less WP-enhanced TOA forcing pattern, but might also lack the WP-enhancement of the surface forcing because the BDC changes differently. Our results are therefore not necessarily applicable to aerosol forcing with pronounced hemispheric asymmetries.

In recent years, much progress has been made to understand how patterns of SST affect radiative fluxes. Especially SST

Green's functions provide a detailed picture about the importance of tropical convective regions for radiative feedbacks. However, less is known about how these SST patterns come about. Simulations with q-flux Green's functions and slab ocean and pacemaker experiments indicate that heat fluxes over the Southern Ocean play an elevated role for SST pattern formation (Lin et al., 2021; Liu et al., 2018a, b, 2022; Hwang et al., 2017; Hu et al., 2022; Kang et al., 2023). Dynamic ocean and atmosphere processes make the temperature pattern time-dependent, even for constant forcings (Heede et al., 2020). Yet, the mechanistic

picture of the connection between forcing pattern and SST pattern is still incomplete. While many pieces are missing on the way to complete this picture, we contribute to filling this gap by identifying relevant processes that cause differences between TOA and surface forcing, emphasizing the relevance of the latter, and by pointing out the atmospheric pathway from TOA forcing to surface forcing to surface temperature pattern specifically for stratospheric aerosol forcing.

*Code and data availability.* The code and data used for this study will be made available upon publication.

**Appendix A: Finding a field of $CO_2$ concentrations that matches the effective TOA forcing of Aerosol**

**A1 Approach**

The goal is to find a field of $CO_2$ concentrations whose effective forcing is equal to the effective forcing field from the stratospheric aerosol ("Aero"), which is known. The idea of the algorithm is to start with an initial guess, compute its effective forcing, and then iteratively increase the $CO_2$ concentration wherever the effective forcing is too negative, and decrease it

wherever the effective forcing is too positive, taking into account the logarithmic dependence of forcing on $CO_2$ concentration.

**A2 Algorithm**

Let $x$ be the $CO_2$ concentration in units of the pre-industrial $CO_2$ concentration (284 ppm), as function of longitude, latitude, and time. The goal is to find a target field $x_t$, whose effective TOA forcing $F$ matches a given target $F_t$. The indices $t, i$ will be used in the following to indicate initial and target fields.

Instantaneous $CO_2$ radiative forcing approximately follows the relationship



$$F \approx c \cdot \log_2(x) \tag{A1}$$

Eq. A1 holds approximately in the global mean with $c \approx 3.7 \ \mathrm{Wm}^{-2}$ (Myhre et al., 1998). However, the instantaneous radiative forcing at each location is determined by the local difference between the temperatures at the surface and the tropopause (Jeevanjee et al., 2021), resulting in a non-uniform TOA instantaneous forcing pattern, even for a uniform change in $CO_2$ con-

centration. Atmospheric adjustments and noise cause further departures of the effective forcing from the instantaneous forcing, both in its pattern and global mean.

From Eq. A1 it follows that

$$F_t/F_i \approx \log_2(x_t)/\log_2(x_i) \tag{A2}$$
$$x_t \approx 2^{F_t/F_i \cdot \log_2(x_i)} \tag{A3}$$
$$x_t = x_i^{F_t/F_i} + \epsilon \tag{A4}$$

(see also Xia and Huang, 2017). In the last step, the approximation symbol is replaced by inclusion of an error term $\epsilon$. The effective forcing field $F$ of any forcing can be computed as the difference of TOA fluxes between a perturbed and an unperturbed simulation with fixed SST and sea ice. Therefore, for any $x_i$, $F_i$ can be determined. Using Eq. A4, one can compute a field $x_t$ that will have the desired forcing field $F_t$, up to an error term $\epsilon$. Using $x_t$ as the new $x_i$, $\epsilon$ can be minimized by repeatedly

applying Eq. A4 to each horizontal grid point. The target $F_t$ remains the same in all iterations.

It is not a priori clear that this algorithm converges. In fact, a few modifications must be made due to errors from adjustments and noise. We apply these modifications in every step.

1. In Eq. A4, $F_t/F_i \to \pm\infty$ for $F_i \to 0$. To address this problem we set $F_i$ in the calculation to $\pm 0.5 \ \mathrm{Wm}^{-2}$ wherever its absolute value is smaller than $0.5 \ \mathrm{Wm}^{-2}$.

2. In our specific case $F_t$ is generally negative, but positive in some places, especially near the poles. In case $F_i < 0 < F_t$, problems arise when $x_i > 1$. From Eq. A1 this is not generally expected ($x_i > 1$ is associated with $F_i > 0$), but can happen due to adjustments or noise. This violation of the assumptions leads to local divergence of the algorithm. One way to fix this is to set $x_t = x_i + c$ wherever $F_i < 0 < F_t \wedge x_i > 1$, instead of applying Eq. A4. We arbitrarily choose $c = 0.5$. The idea is to force the algorithm to increase the $CO_2$ concentration when it is too low in cases where it would

normally decrease it. Similarly, we set $x_t = x_i - c$ wherever $F_i > 0 > F_t \wedge x_i < 1$ to account for the opposite case.

3. After applying Eq. A4, we apply a moving average filter to $\log_2(x)$ with window length of $36°$ longitude and $19°$ latitude in order to smooth the spatial variations.

4. After the moving average filter, we restrict $CO_2$ concentrations to $1/4 < x_t < 4$ in the interest of avoiding too extreme variations.



With the modifications in place, the algorithm converged according to our subjective judgment after three iterations. The resulting field of $CO_2$ concentrations is shown in Fig. 2 (b) of the main manuscript, the effective TOA forcing field is shown in Fig. 3 (a) of the main manuscript. Both clearly share large-scale features, e.g. the most negative forcing and the most strongly reduced $CO_2$ concentration over the WP, but they differ on smaller scales due to adjustments and noise. Monthly-mean fields of effective TOA forcing and $CO_2$ concentrations of $pCO_2$ in comparison to Aero are shown in Figs. A1 and A2.



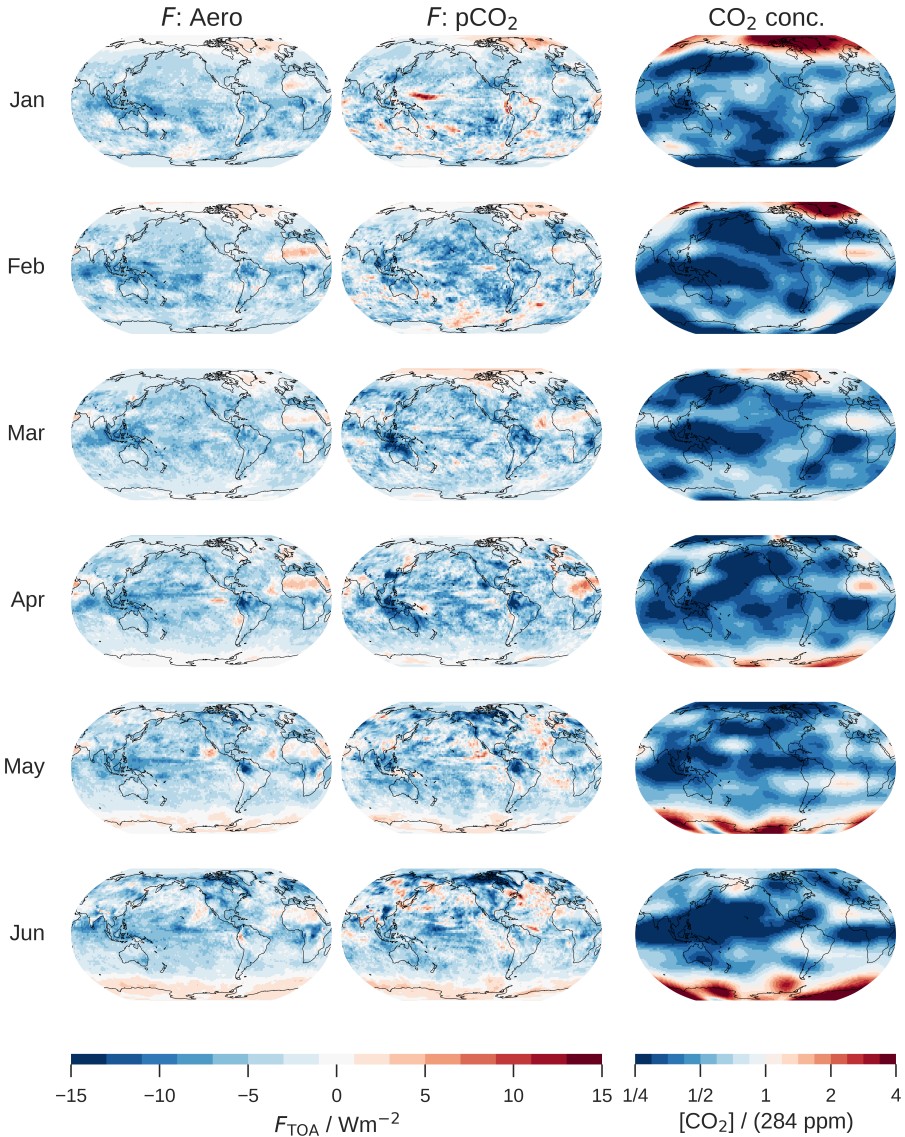

**Figure A1.** Monthly comparison of the effective TOA forcings of Aero and $pCO_2$: January - June. The right column shows the field of $CO_2$ concentrations that results in the effective forcing of the middle column. Note the seasonal dependence of the forcing, with positive forcing over the pole of the winter hemisphere. The seasonal dependence arises despite the time-invariant aerosol profile from the seasonally varying insolation.



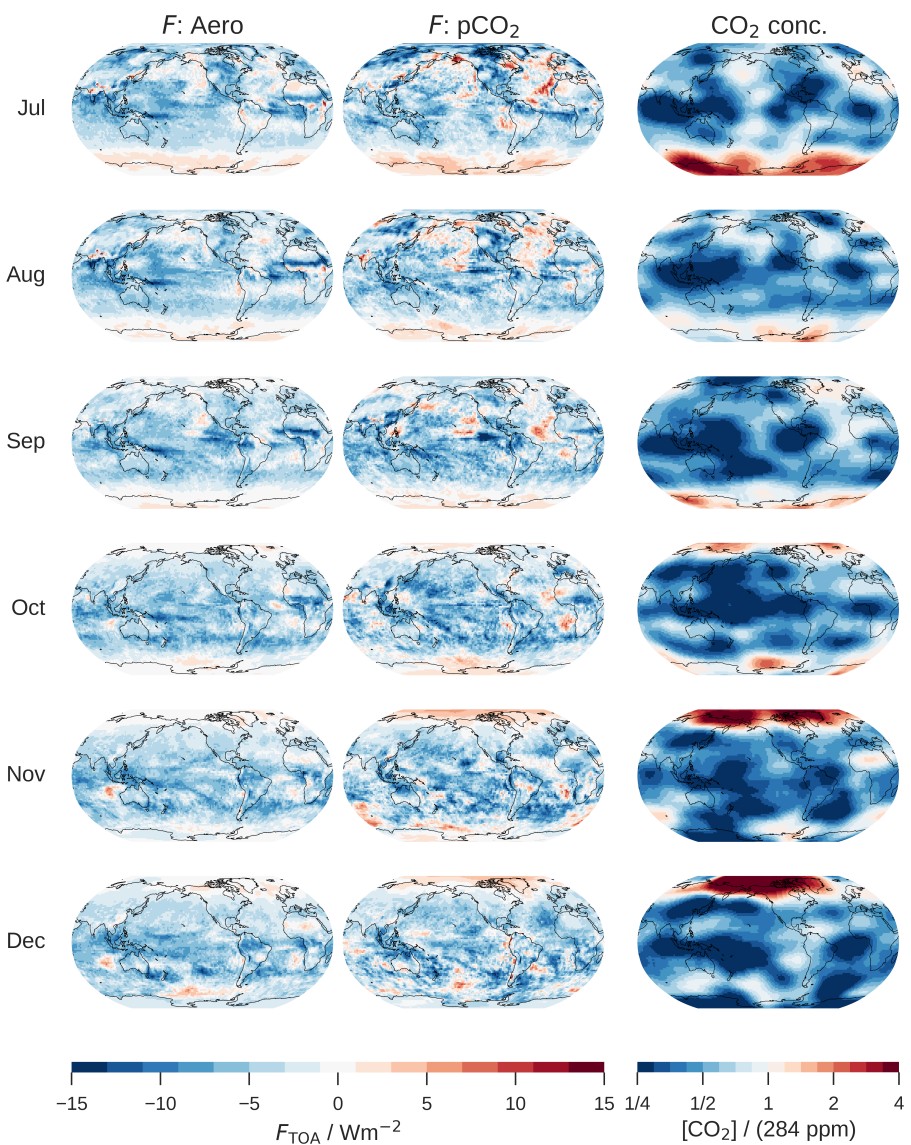

**Figure A2.** Same as Fig. A1, but for July - December





## Appendix B: Tropical-zonal circulation from ocean to land

Although not the main focus of this study, we provide a short explanation why there is a tropical-zonal energy transport from the WP to tropical non-WP regions in all fixed SST simulations (see section 3.3 of the main manuscript).

Fig. B1 shows histograms of variables that indicate an anomalous ocean-land circulation in the fixed SST simulations. Over land, the atmosphere loses energy by radiative and turbulent fluxes almost everywhere, because the land can cool down in the fixed SST simulation, while the ocean can not. This energy loss is compensated by adiabatic heating due to more pronounced downward motion over land.

The energy that is transported to the atmosphere over land is supplied from the atmosphere over ocean, which gains more energy from turbulent and radiative fluxes. The air rises more strongly over ocean, associated with more precipitation and hence convective heating. The circulation must necessarily be closed by movement of air from ocean to land aloft, and from land to ocean near the surface.

The top and middle row show that the WP ocean regions provide proportionally more energy than the tropical non-WP ocean regions, indicated by more positive forcing on the atmosphere and a stronger increase in precipitation. Since the WP is a major region of deep convection, it effectively couples the surface to the free troposphere, which enables energy transport from the surface to the free troposphere and subsequently to the land regions. The prevailing inversion over, e.g., the Eastern Pacific impedes this energy transport. Therefore the majority of the energy transport happens from WP ocean regions to land regions.

This picture is qualitatively similar for the Aero, 0.5 x $CO_2$ and $pCO_2$ simulations, because it does not directly depend on the presence of the forcing agent. Instead, it emerges as a consequence of the air-sea contrast that arises from fixing SST, but not land temperatures. A real-world analog might be the fast response to forcing, where land temperatures react more quickly than SST, such as after volcanic eruptions.

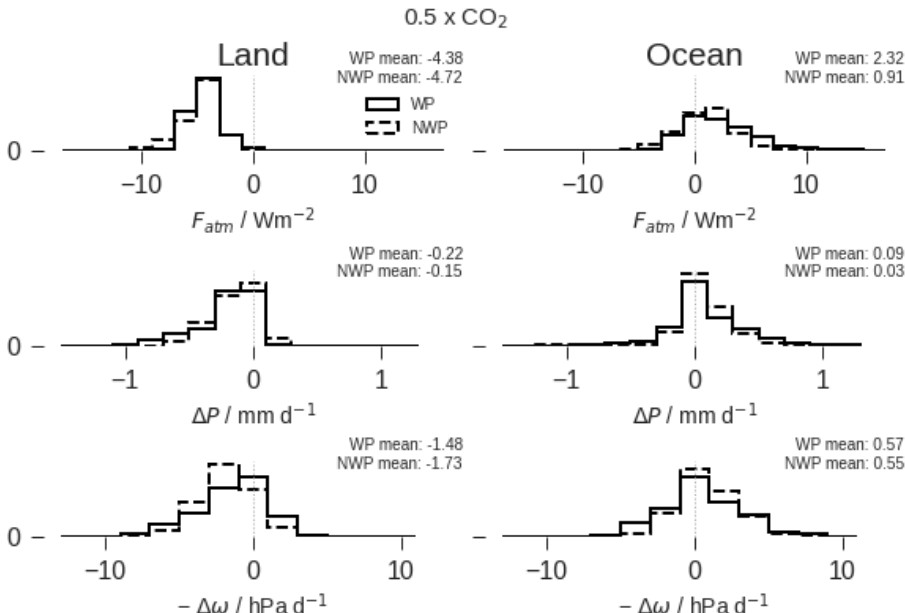

**Figure B1.** Distribution of changes over the tropics, separated into land / ocean and WP / tropical non-WP regions, indicating an ocean-land circulation. Values are taken from the 0.5 x $CO_2$ simulation (Aero and $pCO_2$ qualitatively similar). All histograms show changes of the fixed SST simulation compared to piClim-control. Top: Effective forcing on the atmosphere (= TOA - surface forcing). Middle: Precipitation. Bottom: Negative pressure velocity at 500 hPa (positive values indicate rising motion). NWP = tropical non-WP regions.



## Appendix C: CMIP6 models

|  | Ensemble members |
| --- | --- |
| ACCESS-CM2 | 3 |
| BCC-CSM2-MR | 3 |
| BCC-ESM1 | 3 |
| CESM2-FV2 | 3 |
| CESM2-WACCM-FV2 | 3 |
| CESM2-WACCM | 3 |
| CIESM | 3 |
| CMCC-CM2-SR5 | 1 |
| CNRM-CM6-1-HR | 1 |
| CanESM5 | 65 |
| FGOALS-f3-L | 3 |
| FGOALS-g3 | 6 |
| GFDL-CM4 | 1 |
| GFDL-ESM4 | 3 |
| GISS-E2-1-G | 47 |
| GISS-E2-1-H | 25 |
| INM-CM4-8 | 1 |
| INM-CM5-0 | 10 |
| MIROC6 | 50 |
| NESM3 | 5 |
| NorESM2-MM | 3 |
| SAM0-UNICON | 1 |
| TaiESM1 | 1 |

**Table C1.** CMIP6 models with *hist* simulations included in the CMIP analysis



*Author contributions.* MG and HS conceived this study, building on ideas formulated by HS, CT, and MT. MG performed the simulations, the analysis, and wrote the initial paper draft. All authors discussed the results and revised the paper together.

*Competing interests.* The authors declare that they have no conflict of interest. Matthew Toohey is a member of the editorial board of Atmospheric Chemistry and Physics.

*Acknowledgements.* This work used resources of the Deutsches Klimarechenzentrum (DKRZ) granted by its Scientific Steering Committee (WLA) under project ID mh0066. We acknowledge the World Climate Research Programme, which, through its Working Group on Coupled Modelling, coordinated and promoted CMIP6. We thank the climate modeling groups for producing and making available their model output, the Earth System Grid Federation (ESGF) for archiving the data and providing access, and the multiple funding agencies who support CMIP6 and ESGF. Claudia Timmreck acknowledges funding by the German National funding agency (DFG) research unit FOR 2820: Revisiting The
Volcanic Impact on Atmosphere and Climate-Preparations for the Next Big Volcanic Eruption (VolImpact, Project Number: 545 398006378). We thank Dirk Olonschek for detailed and helpful comments on an earlier version of the manuscript.



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
