# Peer review of "Why does stratospheric aerosol forcing strongly cool the warm pool?"

_EGUsphere, 2024_

## Author Response (AR1)

**Response to Reviewer 1**

Thank you very much for taking the time to give feedback to our manuscript. We have improved the manuscript based on your suggestions, and reply to your comments individually. Especially the comment about the strength of the BDC made us rethink some of our analysis.

**1. Uncertainty intervals should be added to numbers in all the figures and tables.**

We added uncertainties to Table 2 and Fig. 5. Fig. 1 already has errors measures, which are now more clearly visible thanks to Daniele Visioni's suggestions.

We think that including the errors in Figs. 3, 4, and 6 would clutter the figures and make them much harder to read. Instead, we add information on the errors in the captions:

Fig. 3: "Standard errors are negligible compared to the shown precision in (a). In (b), standard errors of the means over the WP and the tropics are $\approx 0.04 - 0.05$ KK-1."

Fig. 4: "The standard errors are generally negligible compared to the shown precision, except: WP mean of surface and TOA–surface: $\approx 0.2$ Wm$^{-2}$, and tropical mean of surface and TOA–surface: $\approx 0.1$ Wm$^{-2}$."

Fig. 6: "In (a) - (f), the standard errors of the means over the WP and the tropics are $\approx 0.04 - 0.05$ KK$^{-1}$. For the differences (g - i), the standard errors are $\approx 0.07$ KK$^{-1}$."

We also added these errors measures to the text whenever we mention the numbers.

**2. One or two figures should be added to explain why there is an enhanced effective radiative cooling at TOA over the warm pool. Contributions from aerosol direct forcings, rapid adjustment of clouds might be displayed.**

We agree that this is information that could be interesting to the reader. We added a figure (see uploaded file) to the appendix, and the following text to the manuscript body:

l. 167: "The TOA forcing of Aero is 1.5 Wm$^{-2}$ more negative in the WP than in the global mean, mainly due to three effects (appendix Fig. C1): first, stronger instantaneous forcing in the tropics than extratropics due to higher aerosol concentration and insolation (− 1 Wm$^{-2}$); second, weaker LW effect over the WP than the whole tropics (− 0.2 Wm$^{-2}$) because the LW effect is weaker over high clouds than over low clouds or the surface; third, more negative SW cloud adjustments over the WP than the whole tropics (− 0.3 Wm$^{-2}$). The last point may be model-dependent and differs from Marshall et al. (2020), who find strong positive SW cloud adjustments in UK-ESM."

**3. Alternative colormap is recommended for Fig. 3(b). Under 0.5xCO2 forcing, there is a cooling everywhere, but it looks as if there is a warming over the southern hemisphere and a cooling in North pole according to the colors of Fig. 3(b).**

We agree that using blue/red can be confusing since it may be mistaken for cooling/warming, when actually the figure shows a slightly different quantity. For the temperature patterns (Figs. 3 b, 6) we switched to a color scheme that is not associated with cooling or warming. We also added notes "strong cooling", "weak cooling", and "warming" to the colorbar for guidance.

**4. The term "non-absorbing" in Table 2 is inaccurate. The absorption cross section of any aerosol particle is greater than zero. "Weak-absorptive" might be used instead.**

The aerosol is prescribed to MPI-ESM with the optical properties extinction, single scattering albedo (ratio of scattering to total extinction), and asymmetry factor (defines forward vs. backward scattering). By setting the single scattering albedo to 1, absorption is completely eliminated. The extinction was multiplied by (1 - initial single scattering albedo) in order not to increase the scattering. This change does not affect the background aerosol. We added this information to the manuscript.

Coming back to your comment, we believe that the term "non-absorbing" is justified, because while there may be no truly non-absorbing aerosol in reality, in the model there is.

**5. The paper emphasizes the importance of BDC intensification, but the strength of BDC has not been quantified. Is there a way to quantify BDC strength, and calculate the percentage change under CO2 and stratospheric aerosol forcings?**

We agree that stating how much the BDC accelerates in our model adds valuable information to the manuscript. We added the following sentence: "A measure for the BDC strength is the tropically averaged residual mean vertical velocity $\bar{w}^*$, which increases in Aero by 10±2 % at 70 hPa and 24±2 % at 30 hPa. In the CO2-forced simulations, these changes are on the order of the uncertainty."

The comment made us rethink our method to calculate the BDC. We came to the conclusion that it would be more exact to use the residual vertical velocity $\bar{w}^*$ (obtained from TEM analysis, see e.g. Butchart 2014: https://dx.doi.org/10.1002/2013RG000448) instead of w. This does not qualitatively affect the results. The additional adiabatic cooling in Aero increases by 10 % when comparing the calculation using $\bar{w}^*$ with the calculation using w. Since our analysis is restricted to the tropics, the influence of the eddies is small, and therefore the difference between $\bar{w}^*$ and w is small. Yet, we changed the manuscript, because this makes the analysis more exact.

We adapted the manuscript (incl. the equations) accordingly, e.g. by adding the following sentence:

l. 148: "The residual mean vertical velocity $\bar{w}^*$ is obtainedfrom a transformed Eulerian mean analysis (e.g. Butchart, 2014). $\bar{w}^*$ combines the mass flux contributions from the mean velocity w and the eddies (Butchart, 2014), and therefore better represents the mass flux than w alone."

**Response to Reviewer 2**

**Thank you for reviweing our manuscript and for giving detailed suggestions for improvement. We have implemented almost all of your suggestions and reply to your comments individually.**

**1. I take no issue with using the differential feedback parameter, but it feels like there is a lack of discussion on what these limited, 10-year, early feedbacks imply and their limitations in terms of understanding longer-term feedbacks and equilibrium temperature changes.**

We added the following sentence to the section about the feedback parameter:

l. 135: "The differential feedback parameter characterizes the transient response to the forcing on a time scale of ten years and bears only very limited implications for the long-term or equilibrium response."

**2. Does the difference BDC changes rely on effective forcing being measured by fixing SSTs but not land temperatures? Is there an argument to be made that we should measure effective forcing with fixed land temperatures?**

The mechanism that accelerates the BDC is mainly based in the stratosphere and therefore does not rely on land temperatures. For this reason, we do not expect the BDC change to rely on the fact that we prescribe ocean, but not land temperatures. Since land-sea temperature contrasts are a source of Rossby waves, we cannot exclude that there is an additional small effect from them on the Brewer-Dobson circulation.

About the second part of the question: There are arguments to be made that land temperatures should be fixed when diagnosing forcing (both at the surface and at the TOA). For a discussion see Forster et al. (2016): https://doi.org/10.1002/2016JD025320

One strong reason against prescribing land temperatures is: "A modified fSST approach has also been tried where both land and sea surface temperatures are fixed to climatological values (Shine et al., 2003). However, this cannot readily be implemented in models with sophisticated land surface schemes that need to capture the diurnal cycle in soil temperatures. (Forster et al. 2016)"

We face the described problem with our model, too. In our interpretation, our study has no substantial arguments to add to Forster et al. 2016.

Technical corrections:

**3. L41-42: typo "acts predominantly SW radiation"**

Thanks, we changed it to "aerosol forcing predominantly affects SW radiation"

**4. WPI_F is used only once, but defined twice (both in 2.4.3 and when it's used).**

We agree that this it seems redundant to define something twice that is only used once. However, we would like to leave it as it is because it seems to be the best alternative. The alternatives are:

a) defining it only when used: possible, but we use it anaologously to the WPI_T, so it makes sense to introduce it at the same point as WPI_T to show their parallelity.

b) not defining it again when it's used: possible, but we believe that it is good to remind the reader at this point of the definition

c) not using the concept at all and instead just referring to it as the ratio of WP to global mean temperature change: possible, but similar as in a), we believe that there is value in using WPI_T and WPI_F in parallel because they describe similar concepts

**5. Figure 5: the green and black circles are a bit hard to pick apart. I'd recommend widening/enlarging the image and markers.**

We agree. We enlarged the image and changed the color to red, so that the points are easier to distinguish.

**Response to Reviewer 2**

**Thank you very much for your encouraging feedback and the suggestions for improvement. We have implemented almost all of your suggestions and reply to each comment individually. We are particularly thankful for the comments about water vapor, which we had not sufficiently discussed.**

**1. Lots of "speculated" in the introduction. I would say that those paper propose specific mechanisms and explanation, rather than mere speculations.**

We changed it so that the sentences now read

l. 38: "The tropically enhanced forcing pattern from aerosols has been **proposed** to be the reason for the pronounced temperature changes in the tropics …"

l. 41: "It has been **argued** that spectral differences could play a role…"

**2. I find Figure 1 rather unclear: I really like schematic figures to introduce papers, but this one I find very confusing. The lack of axis makes the figure seem unmoored, so I'd suggest adding them back. I would also try to separate the result from the standard error by having the former indicated by a larger symbol, and the SE thinner. In the caption, explain better what the x and y axis represent.**

Thanks for the suggestions, we implemented them.

**3. predominantly "on"**

Changed

**4. I wonder why no mention of potential increase of water vapor due to stratospheric heating. I get that the setup of these experiments has fixed chemistry, but TTL warming causing increased H2O concentrations in the LS is also a component to take into account at least in the intro.**

We agree that the effect of water vapor is worth mentioning and added the following sentence to the introduction:

l. 46: "The diabatic heating leads to a cold point warming, which allows more water vapor to enter the stratosphere (Joshi and Shine, 2003; Kroll et al., 2021), with potential impacts on the temperature response (Lee et al., 2023)."

See also our reply to your last point, where we expand on the discussion of water vapor impacts.

**5. Section 2.2.2 Can you say something more about how was the non-absorbing part performed? Did you remove LW absorption only in the stratosphere by reducing the Immaginary part of the coefficient to 0? I think this could be expanded upon, especially as new results were just published with SOCOL doing something similar to isolate the effect of stratospheric sulfur absorption (Wunderlin et al., 2024 https://agupubs.onlinelibrary.wiley.com/doi/full/10.1029/2023GL107285) and it would be**

**interesting to explore whether the techniques are similar, and if there are connections between the results.**

The aerosol is prescribed to MPI-ESM with the optical properties extinction, single scattering albedo (ratio of scattering to total extinction), and asymmetry factor (defines forward vs. backward scattering). By setting the single scattering albedo to 1, absorption is completely eliminated. The extinction was multiplied by (1 - initial single scattering albedo) in order not to increase the scattering. This change does not affect the background aerosol, which is prescribed with a different file. We added this information to the manuscript.

l. 107: "In addition to the fixed SST simulation with the Aero forcing, we perform a simulation with non-absorbing aerosol forcing and only with fixed SST and sea ice, in order to isolate the effects that arise from the stratospheric heating, in particular the acceleration of the BDC (section 3.3). For this simulation we take the forcing from Aero, but set the single scattering albedo (ratio of scattering to total extinction) to one everywhere. The total extinction is then multiplied by (1 - initial single scattering albedo) in order to avoid increases in the reflectivity. For slightly different approaches to isolate the stratospheric heating effects, see Simpson et al. (2019) and Wunderlin et al. (2024). The focus of this study will be on the absorbing aerosol forcing (Aero)."

**6. Figure 3-4: can you zoom in on the maps a bit? They are hard to read and you have space at the margins. The global mean numbers also fall in the map.**

We increased the subfigure size by decreasing the space between them. We moved the global means out of the figures.

**7. Line 208: I don't think it's correct to say you can "redistribute" forcing, considering how forcing is intended. You redistribute the energy, more properly (as you say further below). But just a minor quibble.**

We agree that it sounds a bit strange. However, usually there is no issue with saying that adjustments strengthen or weaken a forcing (e.g. "the positive [rapid adjustments] reduce the volcanic forcing" in Marshall et al. 2020: https://dx.doi.org/10.1029/2020GL090241, or "clouds adjust to the increased CO2 in such a way as to reduce the TOA energy imbalance and so reduce the CO2 radiative forcing" in Andrews et al. 2011: https://dx.doi.org/10.1007/s10712-011-9152-0). So if adjustments can reduce or increase forcing, they can also redistribute it (i.e. increase it somewhere and decrease it somewhere else). Especially when talking about surface and TOA forcing, the word "redistribute" fits well, because the global mean forcing is energetically constrained to be equal at the surface and at the TOA.

We replaced "forcing" with "energy" in one occurrence, but in another one it doesn't work (l. 223): "Taking a perspective from the TOA looking down to the surface, the atmosphere shifts negative forcing from grid points with negative values towards grid points with positive values. Negative forcing is redistributed from columns over land to columns over ocean". It does not make sense to speak of negative energy, and while we could instead say that energy is redistributed from columns over ocean to columns over land, that describes the process less well in our view. Therefore, we would like to adhere to this formulation in this line.

**8. Line 277: you mean to say that the explanation is simply model-dependent, or that you know which specific factor in your model is yielding this conclusions? If the latter, you should**

**say which, or else you should just say "may be different in other models, and would merit a inter-model comparison" (and such experiments or similar ones exist in other models, so you could suggest this easily).**

We agree that the sentence sounds as if we were referring to specifics of our model. Instead, we mean that it arises in fixed SST simulations in general (which only exist in model-world and not in reality). We explain this later in the text (l. 303) with:

"While this circulation arises artificially in our fixed-SST simulations from the fact that land temperatures are not fixed, …"

We changed l. 277 (l. 292 in the revised manuscript) to

"The explanations are somewhat rooted in the way forcing is diagnosed in models, and only partially apply to the real world."

Previously, the sentence read

"The explanations are somewhat rooted in model-specifics and only partially apply to the real world."

We hope that this change makes it clearer.

**9. Figure 5: this could really benefit from a zoom in, I can't tell what I'm looking at. I would also use percentile boxes here instead of standard error.**

Thanks for the suggestions. Reviewer 2 had a similar request regarding image size. We enlarged the image and changed the color of the models with >= 10 realizations to red for better visibility. Following your suggestion, we now report the 20th to 80th percentile interval in the text, in addition to the standard error. We decided to show the standard error to the figure, because it cannot be estimated by eye from the distribution, unlike the 20 - 80 % interval.

l. 339: "The 20 to 80 % intervals are: [-0.22, -0.01] for Krakatau, [-0.25, 0.01] for Pinatubo, [-0.11, 0.05] for the pre-eruption years."

l. 345: "This difference is diminished when only taking into account the models with at least 10 realizations, where the influence of internal variability is reduced. In these models, the transport from the BDC is -0.16 [-0.24, -0.07] W/m^2 for Krakatau and -0.25 [-0.33, -0.16] W/m^2 for Pinatubo, corresponding to roughly 9 % and 14 % of the global mean forcing, respectively."

**10. Line 420: see perhaps also Richter et al., 2017 (https://agupubs.onlinelibrary.wiley.com/doi/10.1002/2017JD026912) about different heights of SO$_2$ injections affecting climate differently, and also Lee et al., 2023 (https://agupubs.onlinelibrary.wiley.com/doi/full/10.1029/2023GL104417) that also discusses the impact of RF efficacy depending on altitude.**

Thank you for pointing out these valuable papers. We cited them, and added the following sentence to the discussion about the limitations due to our fixed ozone profile:

l. 436: "Richter et al. (2017) find slightly higher upper stratospheric upwelling in the simulation without atmospheric chemistry, and almost no tropical temperature differences."

However, we do not follow the argumentation in Lee et al. 2023 about the water vapor **feedback** (in the sense of radiative feedback). Changes in stratospheric water vapor due to the stratospheric heating should be visible within months (since time scales of radiative heating and of the atmospheric tape recorder are ~months) (Kroll et al. 2021: https://doi.org/10.5194/acp-21-6565-2021), and do not depend on surface temperature changes. Therefore, in our opinion, they do not constitute a feedback, but rather an adjustment.

We modified the discussion of the altitude dependence of aerosol forcing. We do not mention the aerosol lifetime effect discussed in Lee et al. (2023) because it does not apply to our study, where aerosol stays forever.

l. 439: "Furthermore, shifting the aerosol profile in altitude or latitude would likely modify the effect on the BDC, so that our results may be dependent on the specific aerosol profile we chose. Aerosol that is injected at greater altitude has been found to cause less negative feedback (Zhao et al., 2021; Lee et al., 2023). Lower injections allow more water vapor to enter the stratosphere, because they more strongly affect cold point temperatures. Lee et al. (2023) argue that this leads to a negative water vapor feedback. Since the increased stratospheric water vapor from cold point heating appears on a time scale of months (Kroll et al., 2021) and independent of surface temperature, we suggest that this does not constitute a feedback, but rather an adjustment. According to our results, the altitude dependence of the feedback could be related to the altitude dependence of the effect of stratospheric heating on the BDC."